# Biological Sensing of Nitric Oxide in Macrophages and Atherosclerosis Using a Ruthenium-Based Sensor

**DOI:** 10.3390/biomedicines10081807

**Published:** 2022-07-27

**Authors:** Achini K. Vidanapathirana, Jarrad M. Goyne, Anna E. Williamson, Benjamin J. Pullen, Pich Chhay, Lauren Sandeman, Julien Bensalem, Timothy J. Sargeant, Randall Grose, Mark J. Crabtree, Run Zhang, Stephen J. Nicholls, Peter J. Psaltis, Christina A. Bursill

**Affiliations:** 1Vascular Research Centre, Lifelong Health Theme, South Australian Health and Medical Research Institute, Adelaide, SA 5000, Australia; achini.vidanapathirana@sa.gov.au (A.K.V.); jarrad.goyne@sahmri.com (J.M.G.); anna.williamson@adelaide.edu.au (A.E.W.); benjamin.pullen@sahmri.com (B.J.P.); pchh7973@uni.sydney.edu.au (P.C.); lauren.sandeman@sahmri.com (L.S.); peter.psaltis@sahmri.com (P.J.P.); 2Australian Research Council (ARC) Centre of Excellence for Nanoscale BioPhotonics (CNBP), Adelaide, SA 5000, Australia; stephen.nicholls@monash.edu; 3School of Medicine, University of Adelaide, Adelaide, SA 5000, Australia; 4Lysosomal Health in Ageing, Hopwood Centre for Neurobiology, Lifelong Health Theme, South Australian Health and Medical Research Institute, Adelaide, SA 5000, Australia; julien.bensalem@sahmri.com (J.B.); tim.sargeant@sahmri.com (T.J.S.); 5Cancer Program, Precision Medicine Theme, South Australian Health and Medical Research Institute, Adelaide, SA 5000, Australia; randall.grose@sahmri.com; 6Division of Cardiovascular Medicine, British Heart Foundation Centre of Research Excellence, Radcliffe Department of Medicine, University of Oxford, Oxford OX3 9DU, UK; mcrab@well.ox.ac.uk; 7Australian Institute for Bioengineering and Nanotechnology (AIBN), The University of Queensland, St. Lucia, QLD 4072, Australia; r.zhang@uq.edu.au; 8MonashHeart, Monash University, Clayton, VIC 3168, Australia; 9Department of Cardiology, Central Adelaide Local Health Network, Adelaide, SA 5000, Australia

**Keywords:** nitric oxide, sensors, macrophages, atherosclerosis

## Abstract

Macrophage-derived nitric oxide (NO) plays a critical role in atherosclerosis and presents as a potential biomarker. We assessed the uptake, distribution, and NO detection capacity of an irreversible, ruthenium-based, fluorescent NO sensor (Ru-NO) in macrophages, plasma, and atherosclerotic plaques. In vitro, incubation of Ru-NO with human THP1 monocytes and THP1-PMA macrophages caused robust uptake, detected by Ru-NO fluorescence using mass-cytometry, confocal microscopy, and flow cytometry. THP1-PMA macrophages had higher Ru-NO uptake (+13%, *p* < 0.05) than THP1 monocytes with increased Ru-NO fluorescence following lipopolysaccharide stimulation (+14%, *p* < 0.05). In mice, intraperitoneal infusion of Ru-NO found Ru-NO uptake was greater in peritoneal CD11b^+^F4/80^+^ macrophages (+61%, *p* < 0.01) than CD11b^+^F4/80^−^ monocytes. Infusion of Ru-NO into *Apoe*^−/−^ mice fed high-cholesterol diet (HCD) revealed Ru-NO fluorescence co-localised with atherosclerotic plaque macrophages. When Ru-NO was added ex vivo to aortic cell suspensions from *Apoe*^−/−^ mice, macrophage-specific uptake of Ru-NO was demonstrated. Ru-NO was added ex vivo to tail-vein blood samples collected monthly from *Apoe*^−/−^ mice on HCD or chow. The plasma Ru-NO fluorescence signal was higher in HCD than chow-fed mice after 12 weeks (37.9%, *p* < 0.05). Finally, Ru-NO was added to plasma from patients (N = 50) following clinically-indicated angiograms. There was lower Ru-NO fluorescence from plasma from patients with myocardial infarction (−30.7%, *p* < 0.01) than those with stable coronary atherosclerosis. In conclusion, Ru-NO is internalised by macrophages in vitro, ex vivo, and in vivo, can be detected in atherosclerotic plaques, and generates measurable changes in fluorescence in murine and human plasma. Ru-NO displays promising utility as a sensor of atherosclerosis.

## 1. Introduction

Nitric oxide (NO) is a gaseous messenger molecule known for its significant regulatory role in almost every cell and it plays a crucial role in maintaining optimum function of the cardiovascular system [1]. NO has a well-established role as a vasodilator produced at nanomolar concentrations in vascular endothelial cells produced by phosphorylation regulated endothelial nitric oxide synthase (eNOS) [2,3]. In addition, activated macrophages within the vascular tissues produce relatively higher levels of NO via activation of inducible nitric oxide synthase (iNOS) [4,5]. Macrophages can be resident or derived from different sources, such as from monocytes or smooth muscle cells, and play a critical role in the development and progression of atherosclerosis [6]. These immune cells can be utilised as a target for detection of the presence of plaque. Derangement of NOS regulation and changes in soluble NO levels in specific tissues are associated with different cardiovascular pathologies, such as hypertension, myocardial infarction, peripheral vascular disease, stroke, and cardiogenic/septic shock [2,7]. In particular, alterations in NO metabolism and presence of reactive nitrogen species are associated with inflammation, a determinant of plaque vulnerability in atherosclerosis [8,9]. Therefore, NO has the potential to be used as marker of early detection of atherosclerosis and to predict the prognosis. Despite such clinical significance, only sub-optimal or surrogate measurements are available as research tools to detect NO in cardiovascular diseases and continue to be used despite reporting variable findings [10,11]. Although methods for detecting NO production by macrophages have been reported [11], they have not been applied to the context of cardiovascular disease.

Light-based methods, such as fluorescence and luminescence detection, represent a more feasible and applicable solution to understand the mechanisms of NO metabolism in biomedical studies, compared to existing radioisotope-based methods. To date, most NO sensor applications have been used in cell-free media conditions, cell lysates or in in vitro cell cultures to demonstrate NO sensor capabilities [11,12]. There are very limited in vivo studies that have tested NO detection probes with non-invasive methods, such as photoacoustic imaging [13]. None of these studies has focused on in vivo detection of NO in atherosclerotic plaque or in blood using fluorescent sensors. Accurate sensing of NO in macrophages and atherosclerosis has the potential to facilitate improved understanding of the role of NO in atherosclerosis and may be applied as a future point-of-care test for the early detection and monitoring of atherosclerosis.

The ruthenium-based NO sensor with the chemical composition of [Ru(bpy)_2_(dabpy)]^2+^ (Ru-NO) is converted to its active form [Ru(bpy)_2_(T-bpy)]^2+^ in the presence of NO, leading to an increase in luminescence [14]. It has previously been validated as an extracellular sensor of secreted NO from endothelial cells [15]. In this study, we aimed to assess the uptake and distribution of the Ru-NO sensor in macrophages in vitro and in in vivo murine models of atherosclerosis as well as to test the utility of using Ru-NO sensor fluorescence to track atherosclerosis in mouse and human plasma. We report that Ru-NO has potential future applications as a research tool to study NO metabolism and macrophage function in atherosclerosis and other cardiovascular diseases.

## 2. Materials and Methods

### 2.1. Ruthenium Based NO Sensor (Ru-NO)

The synthesis and preliminary application of the ruthenium-based NO sensor complex bis(2,2′-bipyridine)(4-(3,4-diaminophenoxy)-2,2′-bipyridine)ruthenium(II)hexafluorophosphate([(Ru(bpy)_2_(dabpy)][PF6]_2_) has been previously described [14]. Serial dilutions from a 100 mM working solution of Ru-NO made in 0.1% dimethyl sulfoxide (DMSO) in phosphate-buffered saline (PBS) were used for in vitro, ex vivo, and in vivo studies with working solutions at a 10–50 µM concentration.

### 2.2. In Vitro Studies 

#### 2.2.1. Human Monocytes and Macrophages

THP1-monocytes were obtained from American Type Culture Collection (ATCC) and grown in RPMI medium with 10% foetal bovine serum (FBS), 1% L-glutamine, and 1% Penicillin-Streptomycin (Sigma-Aldrich, Sydney, Australia). Cells were seeded in a six-well plate at a concentration of 2 × 10^5^ and treated with 200 nM phorbol-12-myristate-13-acetate (PMA) for three days to stimulate differentiation of monocytes to macrophages, as previously described [16]. Macrophages were used two days after changing to PMA-free media. THP1 monocytes and THP1-PMA macrophages were treated with 10 or 50 µM Ru-NO for 24 h for assessments of Ru-NO uptake using flow cytometry, confocal microscopy, and mass cytometry (Appendix A). In parallel, a sample of macrophages were permeabilised using 0.3% saponin in PBS, gently vortexed, and centrifuged at 800× *g* for 10 min. The permeabilised cells were reassessed for the presence/absence of fluorescence signal. THP1 monocytes and THP1-PMA macrophages were also treated in some experiments with 40 µM Ru-NO and 10 µg/mL lipopolysaccharide (LPS) for 18 h [17]. Changes in the Ru-NO fluorescence intensity (λ_ex_ = 488 nm, λ_em_ = 780/60 nm) in treated cells were assessed using flowcytometry. 

#### 2.2.2. Bone Marrow Derived Mouse Macrophages (BMDMs)

BMDMs were isolated from the tibial/femoral bone marrow of C57BL/6J mice, cultured and differentiated [18]. Briefly, the extracted BMDMs were seeded in DMEM F12, 5% FBS, 25 ng/mL macrophage colony-stimulating factor (MCSF), L-glutamine, and penicillin–streptomycin. MCSF (50 ng/mL) was added on Day 5 and granulocyte-MCSF (GMCSF) (50 ng/mL) was added on Day 6 for differentiation. During the differentiation process, the cells were treated with 50 µM Ru-NO for 24 h on Day 8 with or without LPS (100 ng/mL) and interferon gamma (IFNγ, 10 ng/mL) for 16 h. Ru-NO fluorescence within the BMDMs was then analysed using flowcytometry and mass cytometry or used for analysis of iNOS protein expression by Western blotting (Appendix A).

### 2.3. In Vivo Studies in Mice

All animal care and handling procedures were approved by the Animal Ethics Committee of the South Australian Health and Medical Research Institute (SAHMRI, protocol SAM310) and performed in accordance with the Australian Code for the Care and Use of Animals for Scientific Purposes (2013). C57BL/6J and *Apoe*^−/−^ null (homozygous^−/−^) male mice were bred in-house and housed at the SAHMRI Bio-resources animal facility. Food and water were provided *ad libitum*. A pilot study was first conducted to determine the optimum concentration of Ru-NO for in vivo studies and to assess sensor biodistribution and toxicity (Appendix A). At the end of Ru-NO administration, mice were humanely killed using a single administration of 5% isoflurane via inhalation, followed by terminal cardiac puncture and exsanguination.

#### 2.3.1. Peritoneal Macrophages

We tested the uptake and sensor capabilities of Ru-NO in peritoneal macrophages. C57BL/6J male mice of 4–12 weeks were injected with 2.4 µM/kg of Ru-NO intraperitoneally and humanely killed 24 h post-infusion using a single administration of 5% isoflurane via inhalation. Peritoneal cells were collected via peritoneal wash/lavage as previously described [19]. Briefly, ice cold PBS was injected into the peritoneal cavity, massaged gently, and peritoneal fluid collected. These steps were repeated twice to obtain the maximum number of cells. Some cells (3 × 10^4^) were concentrated onto a microscopic slide via a Cytospin™ 4 Cytocentrifuge (Thermo Fisher Scientific, Scoresby, VIC, Australia) and later imaged by confocal microscopy. The remaining peritoneal lavage cells were analysed using flow cytometry to detect Ru-NO fluorescence following incubation with antibodies against myeloid/macrophage markers F4/80 (Australian Biosearch, Karrinyup, WA, Australia) and CD11b (BD Biosciences, Adelaide, Australia) to characterise cell populations.

#### 2.3.2. Murine Model of Atherosclerosis

Apolipoprotein E^−/−^ (*Apoe*^−/−^) mice at 6 weeks of age were fed a high-cholesterol diet (HCD, 21% fat, 0.15% cholesterol) for 12 weeks to develop mid-stage plaques. A cohort of *Apoe*^−/−^ mice that remained on chow for the same time period were included in parallel. Ru-NO (2.4 µM/kg) was injected intravenously 24 h and 5 min before being humanely killed using a single administration of 5% isoflurane via inhalation. Blood was collected through cardiac puncture in EDTA tubes and the aorta, liver, heart, and spleen were harvested and analysed for Ru-NO sensor uptake by flow cytometry. Cross sections of plaque-containing aortic sinuses from optimal cutting temperature compound (OCT, Tissue-Tek)-embedded hearts were fixed post culling and incubated with an antibody against CD68 (1:1000, rat anti-CD68, clone FA-11, #MCA1957GA, Bio-Rad) followed by a fluorescently conjugated secondary antibody (1:2000, Donkey anti-Rat IgG with Alexa Fluor 488, #A-21208) (Invitrogen, Thornton, NSW, Australia) before imaging under confocal microscopy.

### 2.4. Ex Vivo Assessment of Ru-NO Sensor Uptake

#### 2.4.1. Aorta, Spleen, and Liver Cells

*Apoe*^−/−^ mice were fed HCD or normal chow for 16 weeks. Aortas, livers, and spleens were digested (Appendix A), then incubated with PBS or 40 µM Ru-NO (in duplicate) at 37 °C for 60 min. Cells were then washed and analysed using flowcytometry to detect Ru-NO fluorescence following incubation with antibodies against CD11b and F4/80 (myeloid cells/macrophages) and CD31 (endothelial cells).

#### 2.4.2. Blood

From *Apoe*^−/−^ mice fed a HCD or normal diet, 100–150 µL blood was collected via tail bleed, at timepoints of 4, 8, 12, and 16 weeks. Aliquots of 50 µL were immediately added to two tubes containing equal volumes of (1) 40 µM Ru-NO or (2) 40 µM Ru-NO + NO scavenger 2-(4-carboxyphenyl)-4,4,5,5-tetramethylimidazoline-1-oxyl-3-oxide (cPTIO; 200 µM, final concentration [20]), both in PBS. The blood samples were left on ice for 30 min, centrifuged at 7000 rpm for 3 min to isolate plasma then kept on ice before being snap frozen at a consistent time point 100 min after collection. Ru-NO fluorescence was read in thawed plasma samples using a SynergyMx Microplate Reader (BioTek) at λ_ex_ = 450 nm and λ_em_ = 615 nm and the fluorescence values at 615 nm were taken for the analysis. The difference (delta, Δ) between Ru-NO only and Ru-NO + cPTIO fluorescence was calculated to determine the NO-specific signal.

### 2.5. Clinical Blood Samples

Peripheral arterial blood samples were obtained from 50 human subjects who underwent clinically indicated coronary angiography at the Royal Adelaide Hospital, Adelaide, Australia, as approved by the Human Research Ethics Committee of the Central Adelaide Local Health Network, Adelaide, Australia (CALHN HREC # 13579). Informed, written consent was obtained from subjects in accordance with the Declaration of Helsinki and all procedures were performed in accordance with the National Statement on Ethical Conduct in Human Research (2007) in Australia. A 5 mL blood sample was obtained from a radial artery sheath from each patient prior to diagnostic cardiac catheterisation and a 1 mL aliquot of the blood sample was immediately added to a tube containing an equal volume of PBS with the Ru-NO sensor, bringing it to a final concentration of 40 µM of Ru-NO. Another 1 mL aliquot of the same blood sample was also immediately added to a tube containing an equal volume of PBS with the Ru-NO sensor and cPTIO (200 µM, final concentration). Blood added with PBS only and cPTIO only in equal volumes were used as background controls for fluorescence. Following the addition of the sensor/scavenger, the blood samples were left on ice for 30 min, centrifuged at 2600 rpm for 10 min to isolate the plasma, then left on ice before snap freezing one hour after collection. Ru-NO fluorescence was read at λ_ex_ = 450 nm and λ_em_ = 615 nm and the fluorescence values at 615 nm were taken for the analysis.

### 2.6. Statistical Analysis

GraphPad Prism software (GraphPad Software, Inc. La Jolla, San Diego, CA, USA) was used for data analysis. The normality of the distribution was tested using Shapiro–Wilk normality test. The fluorescence values between Ru-NO and PBS treated cells were compared using a paired or unpaired *t*-test as relevant. A one-way ANOVA was used to compare three or more groups with the *p*-values for significant differences derived using a Tukey’s post-hoc test for multiple comparisons. The two-tailed *p*-value for significance was <0.05. All data are reported as mean ± standard error of the mean (SEM).

## 3. Results

### 3.1. Ru-NO Is Detectable In Vitro in THP1 Monocytes and Macrophages 

In vitro, THP1 monocytes and PMA-differentiated macrophages were able to take up the Ru-NO sensor following incubation. This was identified using CyTOF mass spectrometry for detection of elemental Ruthenium; specifically, the ^102^Ru isotype (Figure 1A,B). THP1-PMA macrophages demonstrated approximately two-fold higher Ru-NO sensor uptake compared with THP1 monocytes at both 10 µM (MFI: THP1 monocytes 4.03; THP1-PMA macrophages 9.2, +128%) and 50 µM (MFI: THP1 monocytes 28.12; THP1-PMA macrophages 58.46, +108%). Intracellular Ru-NO fluorescence was also demonstrated in THP1-PMA macrophages using confocal microscopy (Figure 1C). Ru-NO fluorescence was also detected in both THP1 monocytes and THP1-PMA macrophages using flow cytometry (Figure 1D,E). The mean fluorescence intensity (MFI) was greater than six-fold higher in THP1-PMA macrophages compared with THP1 monocytes (THP1 monocytes: 81.1 ± 24.7 vs. THP1-PMA macrophages: 652.6 ± 39.9) and the fluorescence in macrophages was further enhanced by the addition of lipopolysaccharide (+13.8%, *p* < 0.05, Figure 1F). The Ru-NO fluorescence disappeared following permeabilisation, further confirming that the Ru-NO sensor is internalised and the fluorescence signal is specific to the sensor (Figure 1G).

### 3.2. Detection of Ru-NO in Murine Macrophages 

The Ru-NO sensor fluorescence was also detected in CD11b^+^ F4/80^+^ BMDMs from healthy adult C57BL/6 mice (Figure 2A,B). A longer incubation period with the Ru-NO sensor from 1 h to 4 h was found to significantly increase the fluorescence signal. Using Western Blotting, we next confirmed an increase in iNOS protein expression in BMDMs following stimulation with LPS and IFNγ (Figure 2C and Appendix A). Polarised M1- and M2-like macrophages were next tested for their ability to take up the Ru-NO sensor and respond to inflammatory stimuli. In contrast with our findings in human macrophages and despite the induction in iNOS, it was found that stimulation with LPS + IFNγ for 1 h and 4 h reduced the Ru-NO fluorescence signal in the CD86^+^ (M1-like) macrophages (Figure 2D) and CD206^+^ (M2-like) macrophages (Figure 2E). We then determined if the LPS + IFNγ stimulation was affecting cell number or the uptake of the Ru-NO sensor. We found that LPS + IFNγ stimulation increased the proportion of M1-like cells and, conversely, decreased the number of M2-like cells (Figure 2E,F). These effects of LPS + IFNγ were independent of the presence of the Ru-NO sensor, occurring also in the PBS-treated controls. Using mass cytometry, less ^102^Ru was found in BMDMs stimulated with LPS + IFNγ (Figure 2G), indicating an impairment of Ru-NO sensor uptake with inflammatory stimulation.

### 3.3. In Vivo Toxicity, and Detection and Distribution of Ru-NO

A pilot study was conducted in C57BL/6J mice to determine optimum intravenous concentration and exposure time for the Ru-NO sensor. Toxicity testing revealed that infusion of Ru-NO was well tolerated without any adverse events in vivo at 0.6–2.4 µM/kg. Twenty-four hours following administration of 2.4 µM/kg Ru-NO, tissue Ru-NO fluorescence was detected in liver by flow cytometry, greater than for livers of mice infused with PBS (7.1 ± 4.1% of all liver cells, n = 3–5). Ru-NO fluorescence was also present in kidney (+10.7 ± 3.25%) and spleen (+3.2 ± 2.0%), and, to a very modest extent, in aorta (+0.16 ± 0.15%), compared to PBS-infused control mice (Appendix A). Sensor uptake was minimal/negligible in these organs at a dose of 0.6 µM/kg (data not shown). Fluorescence was also negligible in bone marrow, lungs, heart, and plasma after 24 h of either 0.6 and 2.4 µM/kg of Ru-NO infusions (data not shown). Time-course studies using the selected dose of 2.4 µM/kg of Ru-NO sensor injected 5 min, 2, 4, 6, and 24 h before euthanasia revealed that Ru-NO fluorescence was maximum at the 24 h time point in the liver (6.6 ± 4.1%) and kidney (9.5 ± 3.2%), whereas the maximum Ru-NO fluorescence was detected at the 6 h timepoint in the spleen (28.3 ± 15.6) and the aorta (6.4 ± 4.7%) (Appendix A).

### 3.4. Detection of Ru-NO in Peritoneal Macrophages

Confocal microscopy demonstrated an intracellular fluorescence signal in peritoneal macrophages 24 h following intraperitoneal injection of Ru-NO (Figure 3A). When analysed using flow cytometry, we observed a substantial increase in viable fluorescent cells in the Ru-NO exposed peritoneal lavage cells, when compared to PBS injected controls (Figure 3B). There were no differences in overall cell viability or distribution with either PBS or Ru-NO administration to the peritoneum (data not shown). Two distinct populations of cells, (1) myeloid cells (F4/80^−^D11b^+^) and (2) macrophages (F4/80^+^CD11b^+^), were identified in all samples (Figure 3C). When compared to the PBS-treated group (Figure 3D), both CD11b^+^F4/80^−^ myeloid cells and CD11b^+^F4/80^+^ macrophage populations showed robust Ru-NO fluorescence (Figure 3E). The macrophage population demonstrated the highest Ru-NO sensor uptake with consistently higher MFI, when compared to PBS controls (Figure 3F). Within the Ru-NO treated mice, the fluorescence from the Ru-NO sensor was significantly higher in CD11b^+^F4/80^+^ macrophages (MFI: 515.9 ± 121.4; Frequency of Parent, FoP: 61.2 ± 10.6%), demonstrating macrophage-specific uptake of Ru-NO when compared to non-myeloid cells (MFI: 20.5 ± 6.2; FoP: 0.3 ± 0.1%) and myeloid cells (MFI: 202 ± 114.6; FoP: 30.0 ± 9.4%, Figure 3G,H).

### 3.5. Ru-NO in In Vivo Atherosclerotic Plaque Macrophages

We next investigated whether Ru-NO sensor fluorescence was detectable in atherosclerotic plaques in *Apoe*^−/−^ mice fed HCD for 12 weeks. Following infusion of the Ru-NO sensor we were able to detect the presence of Ru-NO fluorescence in atherosclerotic plaques in aortic sinus cryosections by confocal microscopy. Immunofluorescent staining for CD68 revealed that the Ru-NO fluorescence signal localised with CD68^+^ plaque macrophages (Figure 4A–F). Consistent with this, flow cytometry analyses of aortic cell suspensions from these mice found that the aortic CD11b^+^F4/80^+^ macrophage population had a significantly higher in vivo Ru-NO sensor uptake/fluorescence (MFI: 125 ± 33.9; FoP: 65.1 ± 19.1%) compared to CD11b^+^F4/80^−^ myeloid cells (MFI: 6.6 ± 5.8; FoP: 6.5 ± 5.1%) and CD11b^−^F4/80^−^ non-myeloid cells (MFI: 4.3 ± 0.8; FoP: 0.62 ± 0.23% Figure 4G,H). 

### 3.6. Ru-NO in Ex Vivo Cell Suspensions of Atherosclerotic Mice

Ex vivo testing of Ru-NO uptake in cell suspensions from the aorta, liver, and spleen of *Apoe*^−/−^ mice fed on HCD or normal chow for 20 weeks was then conducted using flow cytometry. In the aortic cell suspensions, a distinct CD11b^+^F4/80^+^ macrophage population was identified (Figure 5A) and 36.8% of these cells displayed Ru-NO uptake (Figure 5B). The Ru-NO uptake/fluorescence was then analysed among the myeloid (CD11b^+^F4/80^−^), macrophage (CD11b^+^F4/80^+^) and endothelial cell (CD11b^−^F4/80^−^ CD31^+^) populations. In the mice fed normal chow, the aortic macrophage population displayed the highest fluorescence compared to myeloid cells and endothelial cells (Figure 5C,D). The same Ru-NO uptake/fluorescence patterns were observed in aortic cell suspensions from the HCD fed group (Figure 5E,F). We found a significant increase in the FoP (Figure 5G) and the MFI (Figure 5H) in macrophages from those mice injected with the Ru-NO sensor, compared to the PBS controls in both the HCD and normal chow-fed groups. Interestingly, there were no differences in aortic macrophage Ru-NO fluorescence between normal chow-fed and HCD mice. When we then compared the proportion of aortic macrophages in the chow-fed and HCD mice, it was revealed that the proportion of macrophages in all viable cells was significantly decreased (60.8%) in the HCD group, compared to the normal chow group (Figure 5I). This pattern remained consistent following ex vivo exposure to Ru-NO (Figure 5J).

Comparative assessments for the liver and spleen cells were also conducted (Appendix A). In comparison to the aorta, a relatively higher percentage of macrophages took up the Ru-NO sensor in the spleen (+52.9%) and a relatively lower percentage uptake was reported with the liver cell suspensions (−29.3%). In contrast to the aorta, both spleen and liver cell suspensions contained a higher proportion of macrophages in all viable cells from the HCD group. The uptake patterns in the spleen and liver were, however, similar to the aorta in that the macrophages had the highest uptake/fluorescence of the sensor than myeloid and endothelial cell populations. There remained a lack of change between the Ru-NO sensor uptake between HCD and normal diet fed groups.

### 3.7. Ru-NO in Mouse Blood in Atherosclerosis

We next tested the utility of the Ru-NO sensor to track atherosclerosis progression using blood samples. Serial, four-weekly tail-vein blood samples from atherosclerotic mice were added ex vivo with Ru-NO with/without the NO scavenger cPTIO. Plasma fluorescence was then quantified using spectrophotometry. At every time point, the sample with only Ru-NO reported a significantly higher fluorescence reading compared Ru-NO + cPTIO sample in both HCD and normal diet-fed mice (Figure 6A,B). This indicates the presence of background fluorescence in blood that can be identified by the inclusion of the Ru-NO + cPTIO sample. There was a trend for an increase in Δfluorescence readings (specific NO signal) from the Ru-NO samples with increasing age and time on the HCD (Figure 6C) and chow diets (Figure 6D). When comparing the chow and HCD-fed mice, the Δfluorescence reading was significantly higher in the HCD group (+37.9%, *p* < 0.05) at the 12-week timepoint, when compared to chow fed mice (Figure 6E).

### 3.8. Ru-NO Changes in Clinical Blood Samples

Peripheral arterial blood samples from patients presenting for an angiogram were spiked with Ru-NO with/without NO scavenger cPTIO. Plasma fluorescence was quantified using spectrophotometry. The relevant demographic and clinical details are reported in Appendix A. We identified three categories of patients based on the angiogram findings and clinical manifestations: (1) no or minor coronary artery disease (CAD), <20% narrowing in the vessels, n = 19; (2) stable CAD: >20% narrowing in the vessels without myocardial infarction, n = 20 and; (3) myocardial infarction: >70% narrowing in the vessels with ST elevation myocardial infarction (STEMI) or non-STEMI, n = 11. In all categories, the plasma samples added with Ru-NO only reported a significantly higher reading compared to samples added with Ru-NO + cPTIO, irrespective of the plaque burden or clinical manifestation (Figure 7A). The plasma Δfluorescence reading of the specific NO signal for each patient revealed an upward non-significant trend in the stable CAD group, compared to the no CAD group. Interestingly, the plasma Δfluorescence was significantly lower in the myocardial infarction group, compared to the stable CAD group (−30.7%, *p* < 0.01, Figure 7B). 

## 4. Discussion

In this study, we demonstrated the NO detection capabilities of a Ruthenium-based NO sensor, Ru-NO [14,15], in macrophages, plasma, and atherosclerosis. We show that the Ru-NO sensor is internalised within macrophages and provides NO detection in vitro, ex vivo, and in vivo, including in macrophages in atherosclerotic plaques and aortas. In addition, the Ru-NO sensor was able to detect NO in murine plasma across different stages of atherosclerosis. This was supported by detection of NO in human blood samples from patients with stable and unstable CAD. Using our systematic stepwise approach, we demonstrate bench-to-bedside translation of the Ru-NO sensor, with potential application as a research tool for the measurement of NO in macrophages and as a point-of-care test for atherosclerosis.

The sensitivity and specificity of the Ru-NO sensor was first validated in cell-free media conditions [14] and then in vitro in endothelial cells [15]. It was reported to display comparative responses with commercially available, conventional sensors, such as DAF-FM-diacetate (4-Amino-5-methylamino−2′,7′-difluorofluorescein diacetate) and the Griess assay [15]. The Ru-NO sensor demonstrated relatively higher stability compared to these sensors/assays with no evidence of cytotoxicity at 10–50 µM concentrations in human umbilical vein endothelial cells [15]. We, therefore, used the same concentration range of the Ru-NO sensor in the current study in vitro, ex vivo, and in vivo. In vascular endothelial cells, the Ru-NO sensor was not internalised and, therefore, functioned as an extracellular sensor for the detection of endogenous changes in NO [15]. In contrast, the current study found that the Ru-NO sensor was able to be internalised by macrophages, probably due to their greater phagocytotic capabilities [21].

In our in vitro studies, Ru-NO was internalised by both human THP1 monocytes and THP1-PMA macrophages, as demonstrated by changes in the proportion of cells with Ru-NO fluorescence and confirmed by the absence of these changes with cell permeabilisation. These observations were further supported by confocal microscopy and mass cytometry. Both THP1 monocytes and THP1-PMA macrophages demonstrated concentration-dependent increases in Ru-NO uptake, while THP1-PMA macrophages reported two-fold higher levels of intracellular ^102^Ru compared with THP1 monocytes using mass cytometry. Mass cytometry (CyTOF) detects and quantifies the presence of elemental Ruthenium in the cells, but does not differentiate NO-bound and unbound forms of the sensor. However, our Ru-NO fluorescence data also show higher Ru-NO sensor uptake and/or higher NO levels in THP1-PMA macrophages than THP1 monocytes that was further enhanced in inflammatory conditions. There are several mechanisms that may underly these quantifiable differences in Ru-NO fluorescence. Macrophages are more phagocytic than monocytes [21] and, therefore, may internalise more Ru-NO, which is supported by the CyTOF findings. The Ru-NO sensor only emits a fluorescence signal at specified wavelengths when bound to and activated by NO [14]. This is supported by the PBS controls that had minimal autofluorescence. The Ru-NO fluorescence is, therefore, likely to predominantly represent NO-bound sensor levels inside cells. Previously, in endothelial cells we found similar changes in extracellular NO in response to endogenous stimuli that were compatible with changes in phosphorylated eNOS [15]. Together, these studies support the application of Ru-NO as a sensor to detect both intra and extracellular NO in different cell types associated with vascular function and disease.

Macrophages polarise into M1 and M2-like phenotypes following physiological or pathological stimuli and the M1/M2 distribution within plaques is predictive of the prognosis of atherosclerosis [22]. Increased activity of iNOS-producing NO is evident in M1-like (pro-inflammatory) and some sub-sets of M2-like cells [23]. We found that under non-inflammatory conditions, murine unpolarised (M0) F4/80^+^/CD11b^+^ BMDMs displayed time-dependent increases in Ru-NO fluorescence from 1 h to 4 h. This observation reflects either increased sensor uptake or NO accumulation over time. Following inflammatory stimulation, an expected increase in the proportion of CD86^+^ (M1-type, pro-inflammatory) BMDMs and a decrease in CD206^+^ (M2-type) BMDMs occurred. These changes in macrophage phenotype were independent of the presence or duration of exposure to Ru-NO, suggesting that the Ru-NO sensor does not affect macrophage differentiation or cell viability. Unexpectedly, LPS + IFNγ stimulation reduced Ru-NO uptake/fluorescence in CD86^+^ M1-like BMDMs, despite an increase in the proportion of this cell population. Interestingly, this response was opposite to that of human THP1 macrophages. Previous studies testing different NO sensors have predominately used the murine macrophage RAW264 cell line, which report sensor uptake in undifferentiated macrophages via microscopy. These studies have not demonstrated parallel changes in iNOS activity or reported changes in sensor uptake in the presence of inflammatory stimuli [11,24]. These reports and our observations suggest a cell-species dependent effect in which inflammation has varying effects on Ru-NO uptake into macrophages. There were no differences in Ru-NO fluorescence between the M1 and M2 macrophages. It may have been expected that Ru-NO fluorescence would be higher in the M1-like macrophages, however, variable expression of iNOS in M1, M2 and other sub-types of macrophages has been reported [23] and may explain this lack of change. In summary, our in vitro findings identify Ru-NO as a sensor that can detect NO in macrophages irrespective of cell polarity or inflammation.

In this study we conducted important pilot testing in non-atherosclerotic C57BL6/J mice to determine the biodistribution, optimal dose, time course, and toxicity of the Ru-NO sensor. No adverse toxicological effects were observed. The absence of a Ru-NO fluorescence signal in blood, with low and variable amounts of Ru-NO sensor signal in the liver and kidney 24 h post-systemic infusion, indicates variation in the metabolism and clearance rates of the Ru-NO sensor. This was supported by the time-course studies, in which the highest Ru-NO fluorescence signal occurred in excretory/metabolic organs, such as the kidney and liver at the 24 h time point. In contrast, at the 6 h time point, the highest Ru-NO signal was in the spleen and aorta. This signal declined after 24 h, suggesting there is minimum tissue retention in an uninflamed state in non-atherosclerotic mice. Relatively rapid excretion with minimal tissue retention has significant advantages. Studies with other fluorescence sensors that are retained long-term within tissues following intravenous administration have reported adverse sub-cellular changes [25].

We demonstrated that the Ru-NO sensor can be taken up in vivo into peritoneal macrophages. Peritoneal macrophages from mice injected with the Ru-NO sensor demonstrated a clear shift in fluorescence intensity, compared to PBS controls, suggesting negligible autofluorescence at these wavelengths in vivo. The internalisation of the Ru-NO sensor is a positive attribute as it enables the study of NO using flow cytometry in cell suspensions. Background autofluorescence can, thereby, be kept to a minimum and not be affected by the constituents of culture media for example. Our peritoneal studies found both macrophages and myeloid cells took up the Ru-NO sensor. Consistent with our initial in vitro studies, macrophages had significantly higher uptake/fluorescence of the Ru-NO sensor, compared to myeloid cells and F4/80^−^/CD11b^−^ non-myeloid cells. Once again this could be due to higher levels of iNOS and NO production in macrophages or their greater phagocytic properties. We demonstrate, however, the important utility of the Ru-NO sensor to track macrophage-driven diseases in vivo.

We next tested the Ru-NO sensor in atherosclerotic *Apoe*^−/−^ mice for its ability to detect changes in NO levels in plaque, other organs (liver and spleen) and in blood with increasing development of atherosclerosis. We tested plaque-prone regions of the aorta for Ru-NO sensor uptake. Confocal microscopy on aortic sinus sections revealed a robust signal for Ru-NO sensor fluorescence in plaque that co-localised predominantly with CD68^+^ macrophages. We were able to confirm this observation using flow cytometry on atherosclerotic aortic cell suspensions in which there was also robust aortic macrophage uptake/fluorescence of the Ru-NO sensor following infusion. This signal was far higher than in the non-atherosclerotic C57BL6 mice of the pilot studies, supporting that it has potential utility for detecting disease. While similar findings were found when the Ru-NO sensor was spiked into aortic cell suspensions, Ru-NO fluorescence from aortic macrophages was not different between mice fed the HCD and those on chow. These findings may suggest that the advancement of atherosclerosis in the HCD mice promoted macrophage apoptosis/necrosis, thereby decreasing the number of macrophages able to take up the Ru-NO sensor. This is supported by our analyses that show the total number of viable aortic macrophages on HCD was lower than chow-fed aortas. Endothelial uptake was minimal/negligible compared to monocytes and macrophages. These observations are compatible with our previous findings that were confirmed with ICP-MS analysis [15]. When comparing other NO sensors, endogenous changes in macrophage NO have been previously reported in vitro with a Coumarin-based sensor [24]. In vivo studies in mice with LPS-induced inflammation have used a photoacoustic sensor to image NO in subcutaneous tissue [13]. Intravenous administration of NO sensors has also led to detection in the liver [26]. To our knowledge, the current study is the first demonstration of NO sensing in vivo in plaque macrophages using a NO-specific fluorescence sensor. As demonstrated in previous studies with iNOS imaging [27], NO can, therefore, potentially be utilised as a marker of disease activity. With its specificity, macrophage NO-specific binding, and multimodality imaging potential, Ru-NO presents as a good candidate to detect vulnerable plaque.

As an alternate application, we tested the ability of the Ru-NO sensor to track longitudinal changes in NO levels in serial blood samples from mice fed HCD for 16 weeks and assess if it associated with atherosclerotic disease progression. The mouse plasma samples showed increases in the Δfluorescence NO-specific signal with age and time on HCD/chow. It may have been expected that plasma NO levels from HCD mice would have consistently carried a higher signal than the chow-fed mouse plasma but this was not the case. A significantly higher NO signal did not occur until 12 weeks of HCD and this difference disappeared after 16 weeks of HCD. This suggests there is a certain window in which the Ru-NO sensor can detect changes in atherosclerosis. The chow-fed mice will also develop plaque but after 12 weeks of HCD the plaque is likely to be sufficiently large enough for a detectably different NO signal to be generated from plaque macrophages [28,29]. The lack of change after 16 weeks indicates that the plaques may then have been heading to a stage in which there is significant macrophage apoptosis that lowers iNOS-derived NO production. This hypothesis is consistent with our findings in human plasma from patients with different clinical presentations of CAD. Interestingly, we found the Δfluorescence NO-specific signal in the plasma from myocardial infarction patients was significantly lower than from the plasma of patients with stable CAD. This may reflect that unstable plaque has a substantial level of macrophage apoptosis/necrosis that renders them unable to generate iNOS-derived NO.

Plasma NO can be derived from multiple sources with contributions from both eNOS [3], iNOS [30], nNOS [31], non-enzymatic sources (e.g., dietary nitrites) [32], and ischaemic conditions [30]. Plasma NO concentrations in our study are, therefore, representative of NO production in the whole cardiovascular system from multiple sources in response to the progression of atherosclerosis. It is possible our NO plasma findings in mice and human samples are a reflection of a reduction in eNOS-derived NO. It may also be a manifestation of dysfunctional nitric oxide activity that causes a predisposition to myocardial infarction [33,34]. There is currently no consensus on what NO levels associate with atherosclerosism [35]. Previous studies have reported mixed findings with a decrease in eNOS-mediated expression of NO and reduced nitrite and nitrate levels in HCD mice [36], whereas others have found increased iNOS activity with inflammation [35].

**Limitations:** There are several limitations that may confound the interpretation of the findings of the current study. Detailed distribution studies using radiolabelled Ruthenium were not performed; instead, we used fluorescence as an indicator of the sensor, which would emit a signal only in the presence of NO. Due to the human ethical implications and potential exposure concerns, we could not include Ru-NO in the blood collection syringe, which may have been optimal for a point-of-care test to minimise the time between the blood draw and sensor contact. This limitation is critical when quantifying a volatile molecule such as NO, which could be addressed in future larger clinical studies using this sensor. 

## 5. Conclusions

In conclusion, we report a stepwise in vitro, ex vivo, and in vivo approach for the detection of NO using a molecular sensor. Compared to other multimodality imaging techniques for NO, we identify several positive characteristics of Ru-NO, such as toxicological tolerance at reasonable concentrations, minimal retention in tissues, a bright fluorescent signal for detection with different modalities, and sensitivity to atherosclerosis-related endogenous changes in NO. Due to its molecular size, Ru-NO uptake can be used as a marker of macrophage activity within the vessel wall at the onset and during the progression of atherosclerosis. Most importantly, the application of Ru-NO in blood as a point-of care test has the potential to be further developed, modified, and designed as a clinical tool for early detection of unstable plaque and monitoring of NO-related cardiovascular disease.

## Figures and Tables

**Figure 1 biomedicines-10-01807-f001:**
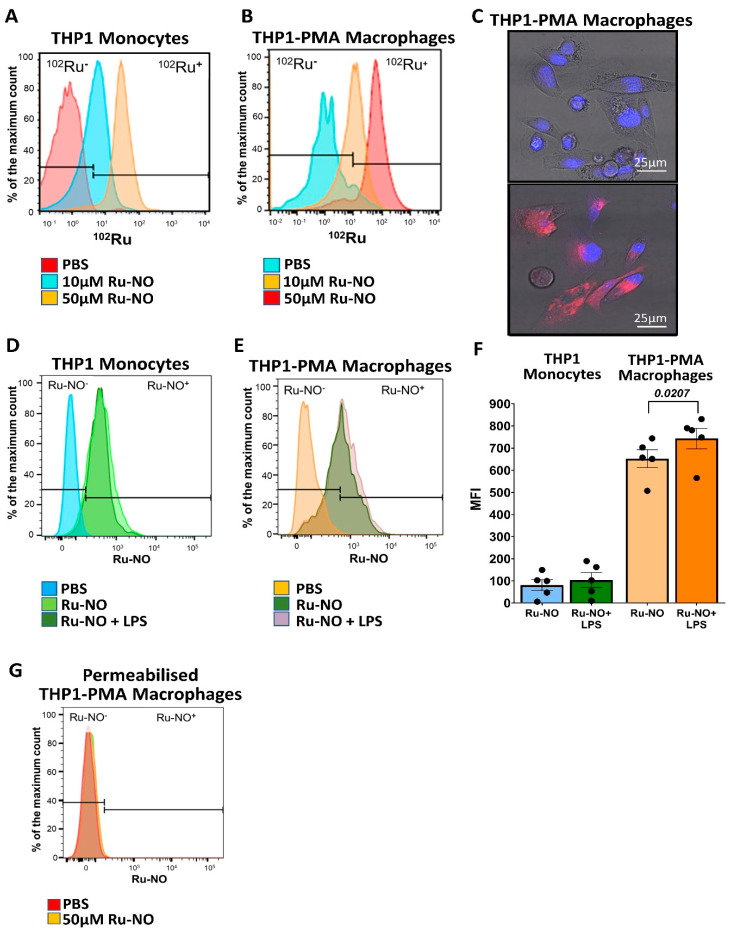
**Ru-NO sensor uptake and detection in human THP1 monocytes and THP1-PMA macrophages.** THP1 monocytes and macrophages were exposed to the Ru-NO sensor to assess sensor uptake and fluorescence detection. The histograms of ^102^Ru in (**A**) THP1 monocytes and (**B**) macrophages exposed to 10 and 50 µM of Ru-NO as assessed by Mass Cytometry (CyTOF). (**C**) Confocal microscopic images of THP1-PMA macrophages treated with PBS (top) and 50 µM Ru-NO (bottom) (red: Ru-NO, and blue: DAPI). Representative flow cytometry histograms of Ru-NO fluorescence in (**D**) THP1 monocytes and (**E**) THP1-PMA macrophages with/without LPS stimulation (**F**) with analyses. (**G**) Lack of Ru-NO fluorescence detection in permeabilised THP1-PMA macrophages by flow cytometry. Mean ± SEM of the mean fluorescence intensity (MFI) of independent experiments, and *p* values derived from a paired *t*-test (n = 5 replicates).

**Figure 2 biomedicines-10-01807-f002:**
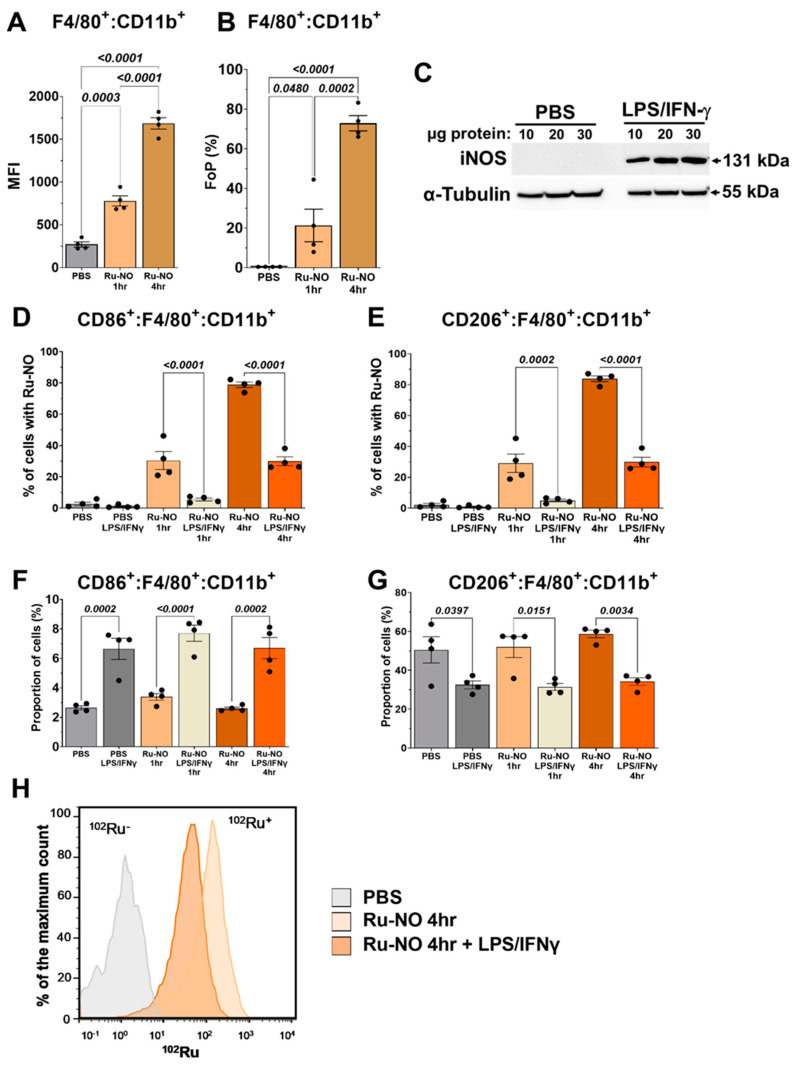
**Ru-NO sensor uptake and detection in murine M1- and M2-like macrophages.** Mouse bone marrow derived macrophages (BMDMs) were assessed for Ru-NO uptake using flowcytometry. (**A**) Mean Fluorescence Intensity (MFI) and (**B**) Frequency of Parental (FoP) of PBS and Ru-NO exposed CD11b^+^F4/80^+^ BMDMs in the absence of inflammation/polarisation. (**C**) Western Blot analysis of iNOS expression in BMDMs with/without LPS + IFN-γ stimulation. The percentage of cells with Ru-NO uptake in (**D**) M1-like and (**E**) M2-like macrophages with/without LPS + IFN-γ. The proportion of (**F**) M1-like (CD86^+^) and (**G**) M2-like (CD206^+^) macrophages with/without LPS + IFN-γ stimulation. (**H**) Mass cytometry histogram for Ruthenium uptake in BMDM with/without LPS + IFN-γ stimulation. Mean ± SEM, *p* values derived from a one-way ANOVA with Tukey post-hoc test (n = 3, performed in quadruplicate) repeated experiments.

**Figure 3 biomedicines-10-01807-f003:**
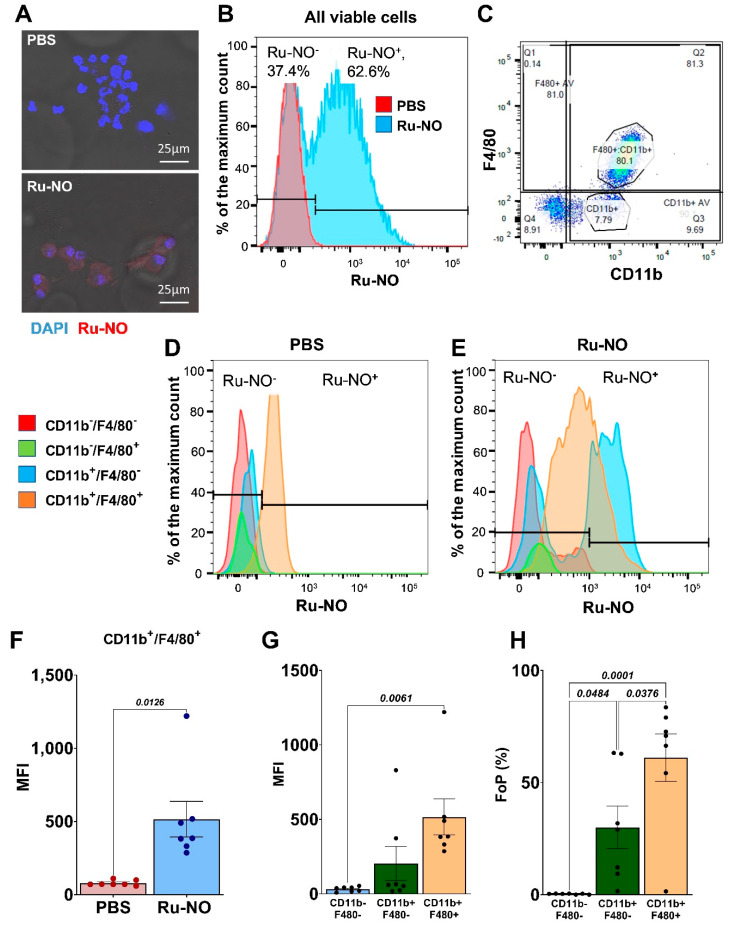
**Ru-NO uptake and detection in peritoneal macrophages** (**A**) Representative confocal microscopy images of Ru-NO intracellular fluorescence signal in peritoneal cells 24 h following intraperitoneal injection of PBS (top) and 40 µM Ru-NO (bottom). (**B**) Representative flow cytometry histograms of peritoneal lavage cells following intraperitoneal infusion of PBS and Ru-NO. (**C**) Representative plot of the distribution of cells with F4/80 and CD11b markers. Representative flow cytometry histograms identifying four populations based on F4/80 and CD11b markers and sensor uptake in (**D**) PBS and (**E**). Ru-NO exposed peritoneal lavage cells. (**F**) Mean fluorescence intensity (MFI) of PBS and Ru-NO exposed CD11b^+^F4/80^+^ peritoneal lavage cells. (**G**) MFI and (**H**) frequency of parental (FoP) analyses of Ru-NO fluorescence in non-myeloid cells (CD11b^−^F4/80^−^), myeloid cells (CD11b^+^F4/80^−^) and macrophages (CD11b^+^F4/80^+^). Mean ± SEM, *p* values derived from a *t* test or one-way ANOVA with Tukey post-hoc test (n = 7 repeated experiments).

**Figure 4 biomedicines-10-01807-f004:**
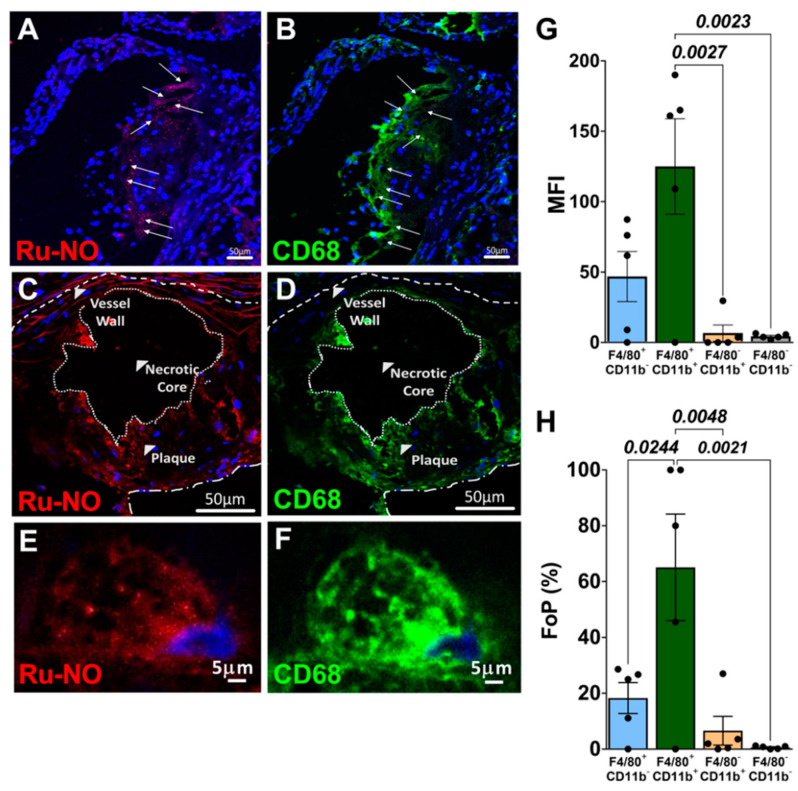
**In vivo detection of Ru-NO in mouse atherosclerotic plaque**. Detection of Ru-NO fluorescence in aortic sinus plaque following intravenous administration of Ru-NO (40 µM) to *Apoe*^−/−^ mice fed a high-cholesterol diet (HCD) for 12 weeks and imaged using confocal microscopy. (**A**,**C**,**E**): Distribution of the Ru-NO (red) in cellular areas of the plaques with nuclear stain DAPI (blue) with increasing magnification. (**B**,**D**,**F**): Comparison with plaque macrophage location (CD68^+^, green). (**G**) Mean fluorescence intensity (MFI) and (**H**). frequency of parental (FoP) of aortic cell suspensions for the detection on Ru-NO fluorescence in non-myeloid cells (CD11b^+^F4/80^−^), myeloid cells (CD11b^+^F4/80^−^) and macrophages (CD11b^+^F4/80^+^) in HCD-fed mice post-Ru-NO infusion. Mean ± SEM, *p* values derived from a repeated measures one-way ANOVA with Tukey post-hoc test (n = 5 animals).

**Figure 5 biomedicines-10-01807-f005:**
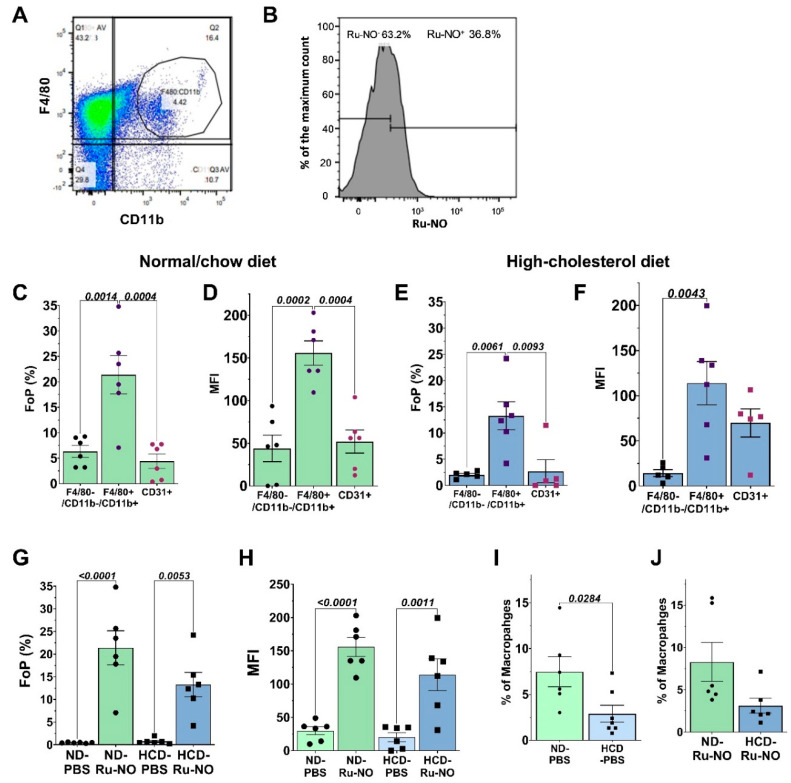
**Ex vivo uptake of Ru-NO by macrophages in aortic cell suspensions from atherosclerotic mice****.** (**A**) Representative plot for the distributions of aortic cell suspensions incubated with antibodies against CD11b and F4/80 markers. (**B**) Representative histogram demonstrating the shift in Ru-NO fluorescence in aortic cell suspensions. (**C**) Frequency of parental (FoP) and (**D**) mean fluorescence intensity (MFI) for Ru-NO fluorescence in myeloid cells (CD11b^+^F4/80^−^), macrophages (CD11b^+^F4/80^+^) and endothelial cells (CD31^+^) in aortic cell suspensions from chow-fed mice. (**E**). FoP and (**F**) MFI for Ru-NO fluorescence in aortic cell suspensions in high-cholesterol diet (HCD)-fed group. (**G**) FoP and (**H**) MFI in CD11b^+^F4/80^+^ macrophages comparing aortic cell suspensions from chow and HCD-fed groups with ex vivo addition of PBS or Ru-NO. Proportion of macrophages in all viable cells that had been incubated with (**I**) PBS and (**J**) Ru-NO. Mean ± SEM, *p* values derived from one-way ANOVA with Tukey post-hoc test for multiple comparisons across different groups (n = 5–6 mice/group).

**Figure 6 biomedicines-10-01807-f006:**
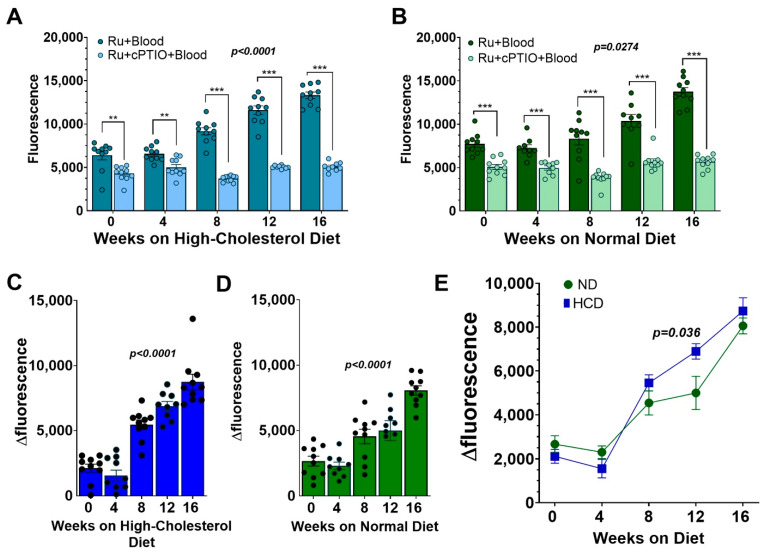
**Plasma Ru-NO fluorescence is higher in mice after 12 weeks of high-cholesterol diet than with chow feeding.** Spectrophotometric readings of Ru-NO fluorescence in plasma from blood samples collected every four weeks added with Ru-NO or Ru-NO + cPTIO (NO scavenger) in mice fed (**A**) high-cholesterol diet (HCD) and (**B**) chow for 16 weeks. Δfluorescence (Ru-NO–Ru-NO + cPTIO) specific NO signal in plasma from mice fed (**C**) HCD and (**D**) chow with (**E**): combined analyses directly comparing the signal between HCD- and chow-fed mice. Mean ± SEM, ** *p* < 0.01 and *** *p* < 0.001 using two-way ANOVA with Tukey post-hoc test for multiple comparisons across different groups. The *p* values in (**A**–**D**) for the linear trend were also calculated (n = 9–10 mice/group).

**Figure 7 biomedicines-10-01807-f007:**
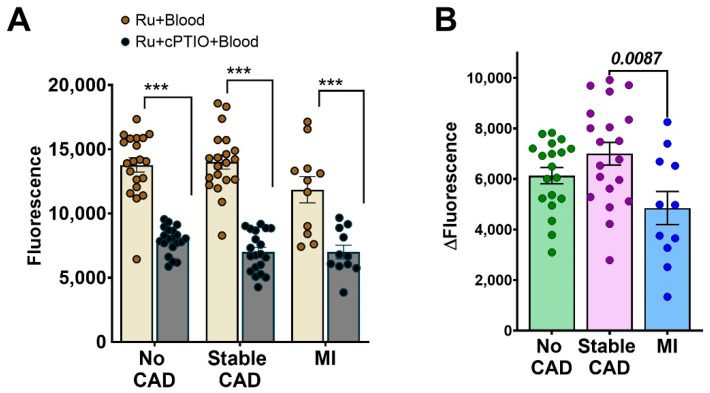
**Ru-NO in blood samples from patients with coronary artery disease.** Plasma Ru-NO fluorescence was quantified using spectrophotometry in peripheral arterial blood samples from patients presenting for an angiogram that were divided into groups of either (1) no or minor coronary artery disease (CAD, <20% narrowing of the vessels, n = 19), (2) stable CAD (<20% narrowing of the vessels without myocardial infarction, n = 20), or (3) with myocardial infarction (MI, >70% narrowing of the vessels with myocardial infarction, n = 11). (**A**) Plasma fluorescence from blood samples added with Ru-NO or Ru-NO + cPTIO (NO scavenger). (**B**) The Δfluorescence (Ru-NO–Ru-NO + cPTIO) NO specific signal from patient blood samples. Mean ± SEM, *** *p* < 0.01 using one-way ANOVA with Tukey post-hoc test for multiple comparisons across different groups.

## Data Availability

Not applicable.

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
