# Peer review of "Biological Sensing of Nitric Oxide in Macrophages and Atherosclerosis Using a Ruthenium-Based Sensor"

_biomedicines, 2022, doi:10.3390/biomedicines10081807_

Round 1

Reviewer 1 Report

This study aimed to investigate the uptake and distribution of the ruthenium-based fluorescent NO sensor (Ru-NO) in macrophages in vitro and in vivo murine models of atherosclerosis (Apolipoprotein E-/- mice; ApoE -/-)and to test the utility of using Ru-NO sensor fluorescence to track atherosclerosis in mouse and human plasma. This study is really interesting, well done with robust methodologies and interesting results clearly described. 

Lines 531-532. "The chow-fed mice will also develop plaque but after 12 weeks of HCD the plaque is likely to be sufficiently large enough for a detectably different NO signal to be generated from plaque macrophages". References are needed for these statements. Please, include the following references regarding the size of atherosclerotic plaques and related NO signal in ApoE -/- mice fed with HFD for 12 weeks. (Giannetto A et al., 2020; Abbate JM et al., 2021). 

Giannetto A, Esposito E, Lanza M, Oliva S, Riolo K, Di Pietro S, Abbate JM, Briguglio G, Cassata G, Cicero L, Macrì F. Protein Hydrolysates from Anchovy (Engraulis encrasicolus) waste: in Vitro and in Vivo Biological Activities. Marine Drugs 2020, 18, 86. 

Abbate JM, Macrì F, Arfuso F, Iaria C, Capparucci F, Anfuso C, Ieni A, Cicero L, Briguglio G, Lanteri G. Anti-atherogenic effect of 10% supplementation of Anchovy (Engraulis encrasicolus) Waste Protein Hydrolysates in ApoE-Deficient Mice. Nutrients 2021, 13, 2137. 

Author Response

Please see file attached.

Reviewer 2 Report

In this work, Vidanapathirana and Goyne et al. test a Ruthenium-based nitric oxide (Ru-NO) sensor in macrophages, plasma, and atherosclerotic plaques. While the manuscript is well-written and results are interesting, major flaws significantly limit the relevance of the presented findings.

Major:

-          How specific is the sensor to detect NO, but not other closely related oxidants such as superoxide anion (O2•-), hydrogen peroxide (H2O2), and peroxynitrite (ONOO-)? This does not seem to have been tested before in the reference provided by the authors (Sci Rep. 2019 Feb 8;9(1):1720) and must be validated.

-          Which is the mechanism of the reaction of NO with the Ru-NO sensor? What is the chemical structure of the Ru-NO sensor and the product after reaction with NO?

-          How does the absorption and emission spectra of the Ru-NO sensor look like? How can the signal be properly discriminated from other fluoresce channels in fluorescence microscopy (Figure 1C) and flow cytometry (Figure 1D)?

-          Monocytes are not CD11b+ F4/80-. According to the gating strategy used by the authors, this population also includes neutrophils (CD11b+ Ly6G+) many other myeloid non-macrophage subsets. In addition, the select populations are not clear in Figure 3C (the “CD11b+ population” also includes CD11b- events).

-          Which is the composition of the diet used for the mice model of atherosclerosis? Is this a HFD (line 159 and elsewhere) or a HCD (line 300 and elsewhere)?

-          The results from plasma Ru-NO fluorescence (Figure 6 and 7) must be validated by using another probe, such as DAF.

-          The title is overstated. It should refer to NO detection in macrophages rather than in atherosclerosis.

Minor:

-          In Figure 1 (and every related method, result and conclusion), “Monocytes” should be replaced by “THP1” and “Macrophages” by “THP1+PMA”.

-          Autofluorescence signal must be included in Figure 5B.

Author Response

Please see file attached.

Round 2

Reviewer 2 Report

The manuscript have greatly improved from its original version.

References are still needed for NO release from macrophages upon inflammatory activation (line 50) and for the use of cPTIO as NO scavenger (line 178). Please include Basic Res Cardiol. 2019 Aug 19;114(5):38 and Ecotoxicol Environ Saf. 2020 Dec 1;205:111186, respectively.

Otherwise, I have nor further concerns.

Author Response

Reviewer 2 comments: The manuscript have greatly improved from its original version.

References are still needed for NO release from macrophages upon inflammatory activation (line 50) and for the use of cPTIO as NO scavenger (line 178). Please include Basic Res Cardiol. 2019 Aug 19;114(5):38 and Ecotoxicol Environ Saf. 2020 Dec 1;205:111186, respectively.

Response: The two references recommended by the Reviewer have now been added to the manuscript (See, in red, pages 18 and 19).